# Application of Multi-Source Data for Mapping Plantation Based on Random Forest Algorithm in North China

**Fan Wu** [1,2], **Yufen Ren** [1,3] and **Xiaoke Wang** [1,*]

1    State Key Laboratory of Urban and Regional Ecology, Research Center for Eco-Environmental Sciences, Chinese Academy of Sciences, Beijing 100085, China
2    University of Chinese Academy of Sciences, Beijing 100049, China
3    Beijing Urban Ecosystem Research Station, Research Center for Eco-Environmental Sciences, Chinese Academy of Sciences, Beijing 100085, China
*    Correspondence: wangxk@rcees.ac.cn

**Abstract:** The expansion of plantation poses new challenges for mapping forest, especially in mountainous regions. Using multi-source data, this study explored the capability of the random forest (RF) algorithm for the extraction and mapping of five forest types located in Yanqing, north China. The Google Earth imagery, forest inventory data, GaoFen-1 wide-field-of-view (GF-1 WFV) images and DEM were applied for obtaining 125 features in total. The recursive feature elimination (RFE) method selected 32 features for mapping five forest types. The results attained overall accuracy of 87.06%, with a Kappa coefficient of 0.833. The mean decrease accuracy (MDA) reveals that the DEM, LAI and EVI in winter and three texture features (entropy, variance and mean) make great contributions to forest classification. The texture features from the NIR band are important, while the other texture features have little contribution. This study has demonstrated the potential of applying multi-source data based on RF algorithm for extracting and mapping plantation forest in north China.

**Keywords:** plantation; forest classification; random forest; feature importance; multi-source data





## 1. Introduction

To meet the increasing demand for timber and implement regional ecosystem restoration, China has developed large areas of plantation since the 1960s [1]. According to the forest inventory data [2], the area of plantation in China has passed 7.5 million hectares, accounting for more than one third of the country's total forest area. Accurate forest mapping provides important support for scientific management and sustainable utilization of forest resources. Nevertheless, due to the various plantation tree species and long periods of development, the spatial distribution pattern of plantations remains unclear in China, especially in northern areas.

Remote sensing is particularly effective for obtaining the spatial distribution of forests, as it provides great coverage with high levels of detail compared to traditional survey and inventory techniques [3–6]. Nevertheless, for plantation extraction, single-season or single-scene remote sensing images may be insufficient due to the lack of characteristics. Single-season or single-scene images only reflect the spectral characteristics of the forest at the time of photographing and cannot capture the spectral information of forests in different seasons [7]. Furthermore, due to the limitations of spatial resolution and the number of spectral bands, low separability and classification confusions may occur based solely on spectral features [8]. Multi-source data contain not only remote sensing data, but also auxiliary data helpful for classification. Multi-source data provide different types of information in forest classification, such as multi-spatial resolution images, surface reflectance of multiple seasons, phenological characteristics and textural information [9,10]. The use of multi-source data can be divided into three groups: multi-temporal data, multi-spatial resolution data, and multi-platform data. Multi-temporal data capture the unique

phenological characteristics of the target forest type by multi-temporal or time series data [11]. For example, a rubber plantation has an obvious defoliation period different from natural evergreen forests in some regions. Using multi-temporal imagery, it is easy to obtain the phenological characteristics of a rubber plantation and achieve high accuracy for mapping the rubber plantation [12]. Multi-spatial resolution data application provides complementary information, and improves the resolution and clarity of images [13]. Multi-platform data application focuses on a combination of remote sensing data and other auxiliary data. For plantations, the spatial distribution might be closely related to various factors such as topography, climate, soil, and afforestation system [14]. Therefore, besides remote sensing data, non-remote sensing data are also used as vital auxiliary information to participate in the extraction of plantations to achieve desirable accuracy [15]. Auxiliary information includes forestry statistics data, soil data, afforestation standards or procedures, as well as experts' knowledge and experience [16].

Compared with using spectral bands only, multi-source data carry more information and increase the separability of forest types [17–19]. Multi-source data application has been conducted in processes of mapping a Larch plantation [20], rubber plantations [21,22] and a *Eucalyptus* plantation [23]. Although this framework has obtained increasing attention, it is still challenging to distinguish the plantation and natural forest in some regions because of their similar phenological characteristics and mixed tree species composition. In addition, there are many combinations of features when using multi-source data. If the features generated by all possible combinations are involved in the classification, it may not only require a huge amount of calculation, but also cause a dimensionality problem [24]. For researchers, how to choose variables and their corresponding input parameters to obtain the ideal results is a key question well worth exploring. Although many researchers have noticed this issue and discussed the feature selection [25,26], most studies apply feature combinations according to past experience or apply all features directly.

There are four types of features obtained from multi-source data: spectral features, vegetation index (VI), texture features and topography features. A spectral feature is the most widely used and basic identification feature of forest classification by remote sensing [27]. By calculating the mean, variance, standard deviation or dispersion of the reflectance, quantitative analysis can be established. Vegetation index is a feature commonly used in remote sensing research [28–31]. Vegetation has different absorption and reflectance in the spectral bands. Vegetation index generates new features by combing multiple spectral bands, as a quantitative measure of vegetation vitality, and captures phenological characteristics of different forest types [32,33]. The application of vegetation index can improve the interpretation ability of images and improve classification accuracy. Li et al. [34] discussed the effect of different vegetation indices obtained by Landsat Thematic Mapper imagery, and selected the best index suitable for the study area, which improved the vegetation classification accuracy. Kun et al. [35] fused Landsat ETM+ data and time series normalized difference vegetation index (NDVI) to obtain fine-resolution NDVI data for improving forest cover classification accuracy. Senf et al. [36] used time series NDVI and enhanced vegetation index (EVI) for mapping a rubber plantation in southwest China. VI is generally combined with spectral bands and texture features for forest classification.

Image texture, which represents the visual spatial variability, has been widely utilized in identifying forest types [37–39]. Various texture measures aiming to quantify the brightness, smoothness, regularity, etc., of an image have been developed based on the gray level co-occurrence matrix (GLCM) [40]. Usually, plantation trees are clearly outlined and more orderly compared with natural forest on high-resolution images. The plantation patches can be distinguished easily by visual interpretation. This characteristic is beneficial to the plantation extraction. Therefore, many scholars have carried out research studies on forest type classification or tree species extraction by adding texture features to quantify texture characteristics. Franklin et al. [41] used high spatial resolution aerial photographs (0.3–1 m) to classify tree species, and added two GLCM textures (homogeneity and entropy), which improved the classification accuracy by 10–15%. Johansen et al. [42] used QuickBird data to

classify forest and found that the addition of texture information improves the classification accuracy by 2–19%.

Additionally, the topography feature is significant for forest classification. Studies show a correlation between the spatial distribution of forest types and topography information [43]. In rugged regions, topographic variables are usually included in forest classification because topographic variation would influence the spatial distribution of forest types due to its influence on temperature and moisture gradients [44]. Furthermore, topography influences afforestation practices. For instance, afforestation is impossible on extremely steep slopes and it is more prevalent on flat areas [45]. Topographic information can be represented by a digital elevation model (DEM), slope and aspect [46]. After adding topographic information, the classification accuracy can be significantly improved. Strahler et al. [47] demonstrated a 27% increase in accuracy in tree species classification by adding topographic information. Other studies found that the addition of topographic feature can improve the overall accuracy by 5–10% of forest classification [6,48].

Random forest (RF) is an ensemble learning algorithm that has been documented as an excellent performer for the analyses of many complex remote sensing datasets [49–51]. It exhibits many desirable properties, including high accuracy, processing thousands of input variables and integrated measures of variable importance [52,53]. Moreover, RF is robust against outliers and noise [54]. Moreover, the parameter setting for RF is simple and the computation process is fast. RF has demonstrated excellent performance for land classification studies [55–59].

GaoFen-1 (GF-1), the first satellite of China's high-resolution Earth Observation System, was launched on 26 April 2013. GF-1 has 16 m resolution multispectral WFV cameras. The camera has four multispectral bands spanning from the visible to the near infra-red [60]. Images captured by GF-1 WFV with rich spectral information and broad spatial coverage make it a suitable source for mapping and monitoring forest.

In this study, multi-source data, including GF-1 WFV imagery, high-resolution imagery from Google Earth Pro, DEM and forest inventory data were combined for mapping plantations and natural forest in Yanqing district, Beijing, China. The specific objectives of this study are to (1) define appropriate parameters for RF models; (2) obtain an optimal subset of features for RF models and identify the effective features; and (3) discuss the applicability of multi-source data on the mapping plantation in mountainous areas.

## 2. Materials and Methods

### 2.1. Study Area

Yanqing is located northwest of Beijing at 40°16′–40°47′N and 115°44′–116°34′E. The total area is 199,375 ha, of which the mountainous area accounts for 72.8%. It belongs to the continental monsoon climate with much rainfall from June to September and cold in winter. The annual average temperature ranges from 2 °C to 14 °C and annual precipitation averages 460 mm. Since the 1990s, afforestation projects have been launched in Yanqing including natural forest reservation, Grain for Green Project, and the plain afforestation. The forests in Yanqing are mostly plantation and natural secondary forests, with a total area of 112,262 ha, accounting for 56.3% of the whole area. The forest species of natural forest in this district is *Quercus mongolica* Fischer ex Ledebour with a small amount of *Pinus tabuliformis* Carr. The major forest species of plantations in this district are *Platycladus orientalis* (Linn.) Franco, *Pinus tabuliformis* Carr., and *Populus davidiana*. *Platycladus orientalis* (Linn.) Franco and *Pinus tabuliformis Carr.* are evergreen coniferous forests. *Quercus mongolica* Fischer ex Ledebour and *Populus davidiana* are deciduous broad-leaved forests. Figure 1 shows forest of Yanqing and forest type samples used in our study.

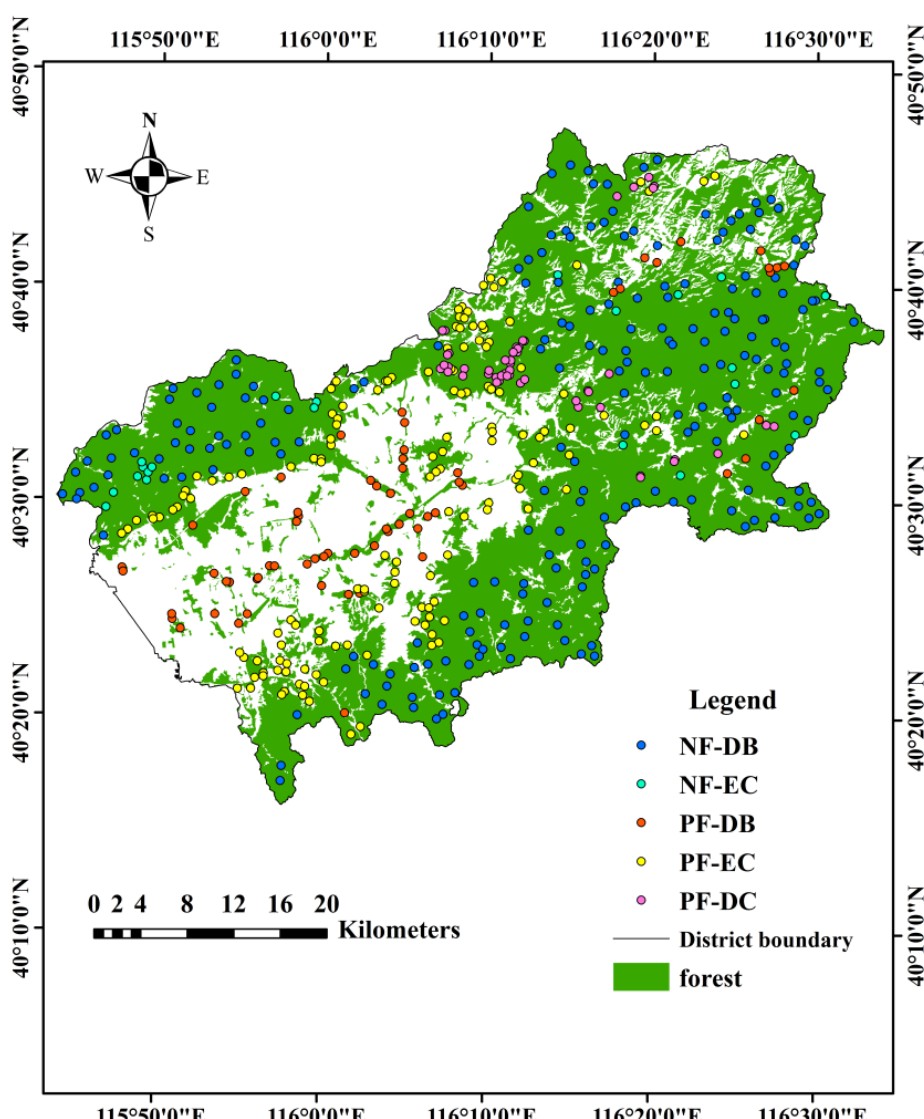

**Figure 1.** Study area and forest type samples NF-DB: natural deciduous broad-leaved forest; NF-EC: natural evergreen coniferous forest; PF-DB: deciduous broad-leaved plantation; PF-EC: evergreen coniferous plantation; PF-DC: deciduous coniferous plantation.

*2.2. Data Acquisition and Pre-Processing*

We acquired Gaofen-1 Wide-Field-of-View (GF-1 WFV) images from the China Center for Resource Satellite Data and Application (http://www.cresda.com accessed on 15 June 2020). Two images (path/row = 3/87) with four bands (band1 blue: 0.45–0.52 μm, band2 green: 0.52–0.59 μm, band3 red: 0.63–0.69 μm, and band4 near-infrared: 0.77–0.89 μm) at 16-m spatial resolution were obtained on 26 September 2015 and 12 November 2015. Both images are cloud-free. Pre-processing steps were performed in ENVI 5.3.1 software by Harris Corporation in Melbourne, Florida, United States, including radiometric calibration, atmospheric correction, orthorectification and projection definition. The FLAASH (fast line-of-sight atmospheric analysis of spectral hypercubes) model was used for atmospheric correction. The orthorectification was performed by rational polynomial coefficient (RPC) files.

The digital elevation model (DEM) at 30-m spatial resolution was obtained from Resource and Environment Science and Data, Institute of Geographic Sciences and Natural Resources Research, Chinese Academy of Sciences (http://www.resdc.cn/ accessed on 20 June 2020).

To obtain training samples for the RF classification and validation samples for accuracy assessment, we collected sample points from a forest inventory map of Yanqing and high-resolution satellite imagery in Google Earth Pro. We obtained the forest inventory map of Yanqing from the National Forestry and Grassland Data Center (http://www.forestdata.cn/ accessed on 6 July 2020). We obtained 670 sample points including 313 natural forest sample points and 357 plantation sample points (Table 1). All samples were randomly divided 70%/30% for training and validation, respectively.

**Table 1.** The number of samples.

| Forest Type | | Abbreviation | Number of Samples |
|---|---|---|---|
| Natural forest | deciduous broad-leaved | NF-DB | 222 |
| | evergreen coniferous | NF-EC | 91 |
| Plantation | deciduous broad-leaved | PF-DB | 108 |
| | evergreen coniferous | PF-EC | 150 |
| | deciduous coniferous | PF-DC | 99 |

*2.3. Feature Extraction and Selection*

2.3.1. Feature Extraction

Four types of features were considered in our study: spectral features, vegetation indices, texture features, and topography features. Four bands of GF-1 WFV in two seasons were taken as the spectral features. We generated nine vegetation indices from GF-1 WFV images including difference vegetation index (*DVI*), enhanced vegetation index (*EVI*), green difference vegetation index (*GDVI*), green normalized difference vegetation index (*GNDVI*), green ratio vegetation index (*GRVI*), leaf area index (*LAI*), normalized difference vegetation index (*NDVI*), simple ratio vegetation index (*SR*) and soil-adjusted vegetation index (*SAVI*). The equations are as follows:

$$DVI = \rho_{NIR} - \rho_{red} \tag{1}$$

$$EVI = 2.5 \times \frac{(\rho_{NIR} - \rho_{red})}{(\rho_{NIR} + 6 \times \rho_{red} - 7.5 \times \rho_{blue} + 1)} \tag{2}$$

$$GDVI = \rho_{NIR} - \rho_{green} \tag{3}$$

$$GNDVI = \frac{\rho_{NIR} - \rho_{green}}{\rho_{NIR} + \rho_{green}} \tag{4}$$

$$GRVI = \frac{\rho_{NIR}}{\rho_{green}} \tag{5}$$

$$NDVI = \frac{\rho_{NIR} - \rho_{red}}{\rho_{NIR} + \rho_{red}} \tag{6}$$

$$SR = \frac{\rho_{NIR}}{\rho_{red}} \tag{7}$$

$$SAVI = \frac{1.5 \times (\rho_{NIR} - \rho_{red})}{(\rho_{NIR} + \rho_{red} + 0.5)} \tag{8}$$

where $\rho_{NIR}$, $\rho_{red}$, $\rho_{green}$, and $\rho_{blue}$ are the surface reflectance values of near infrared, red, green, and blue bands of GF-1 WFV images. LAI data is from a moderate resolution imaging spectroradiometer (MODIS) MOD15A2H Version 6 product (https://ladsweb.modaps.eosdis.nasa.gov/ accessed on 6 July 2020) [61].

In this study, texture features are extracted based on gray level co-occurrence matrix (GLCM). It describes the spatial relationship and structural characteristics of image gray value proposed by Haralick in 1973 [62]. Eight texture features including mean (ME), variance (VAR), homogeneity (HOM), contrast (CON), dissimilarity (DIS), entropy (ENT), second moment (SM), and correlation (COR) were computed. The window size is very

important in extraction process of texture features [63,64]. Too large a window size may cause insufficient texture information, high misrecognition rate of each category or systematic errors. Too small a window size may lead to low separability for classification or excessive extraction time. Considering the spatial resolution of remote sensing data (16 m), the diameter of forest patches (90–200 m) [65], the crown width of trees (1–7 m), texture features were retrieved with three window sizes in $3 \times 3$ ($48 \times 48$ m), $5 \times 5$ ($80 \times 80$ m) and $7 \times 7$ ($112 \times 112$ m) in this study.

Topography features include DEM, slope and aspect, the latter two are extracted from the DEM at 30 m.

### 2.3.2. Feature Selection

In Section 2.3.1, we obtained a total of 125 features derived from GF-1 and DEM imagery, including 8 spectral features, 16 vegetation indices, 96 texture features and 3 topography features. Before classification, the number of features is usually reduced and some key features are selected to reduce the complexity of the model, speed up the construction of the model, and reduce the occurrence of overfitting problems [66–68]. In this study, we applied the recursive feature elimination (RFE) method to select features using "caret" package in R language version 4.1.2. First, the method applies all features and ranks the feature importance. The feature importance is ranked by measuring mean decrease accuracy (MDA) using random forest model (rfFuncs in "caret" package) [69] and we obtain the overall accuracy of random forest models. Then, RFE deletes the features with the lowest importance one by one to determine the overall accuracy under different feature numbers [70,71]. Finally, a subset that achieves the maximum overall accuracy and the minimum number of features are obtained.

### 2.4. RF Classification

We used random forest (RF) classifier supported by R language version 4.1.2 to map five forest types in this study. RF is a learning method that consists of many decision trees. Each tree is built by bootstrapping, a sampling method with replacement [72]. This method has the advantage of handling hundreds of input variables and is less prone to overfitting. In addition, this method can rank feature importance for classification. Two crucial parameters were set in RF classifier: the number of decision trees (ntree) and the number of features for each node (mtry). The default of mtry is set up to the square root of the total number of input features. In this study, to create suitable RF models for forest type mapping, RF models were built with different combinations of mtry and the maximum ntrees (500) using the training samples. The feature importance was ranked in RF algorithm by MDA (mean decrease accuracy). The MDA computes the difference in prediction error before and after permutation of each feature. The larger of MDA value, the more important of the feature. In the accuracy assessment process, along with overall accuracy, Kappa coefficient, producer's accuracy and user's accuracy, area accuracy is used for comparing classification results with inventory data. The equation is as follows:

$$K_i = \left(1 - \frac{|A_i - A_0|}{A_0}\right) \times 100\% \tag{9}$$

where $K_i$ is the area accuracy of forest type $i$, $A_i$ is the area of forest type $i$ from classification results, and $A_0$ is the area of forest type $i$ from inventory data. We used "randomForest" package in R language version 4.1.2 for classification. Figure 2. is the methodological flowchart illustrating the important steps to produce the forest map in the study.

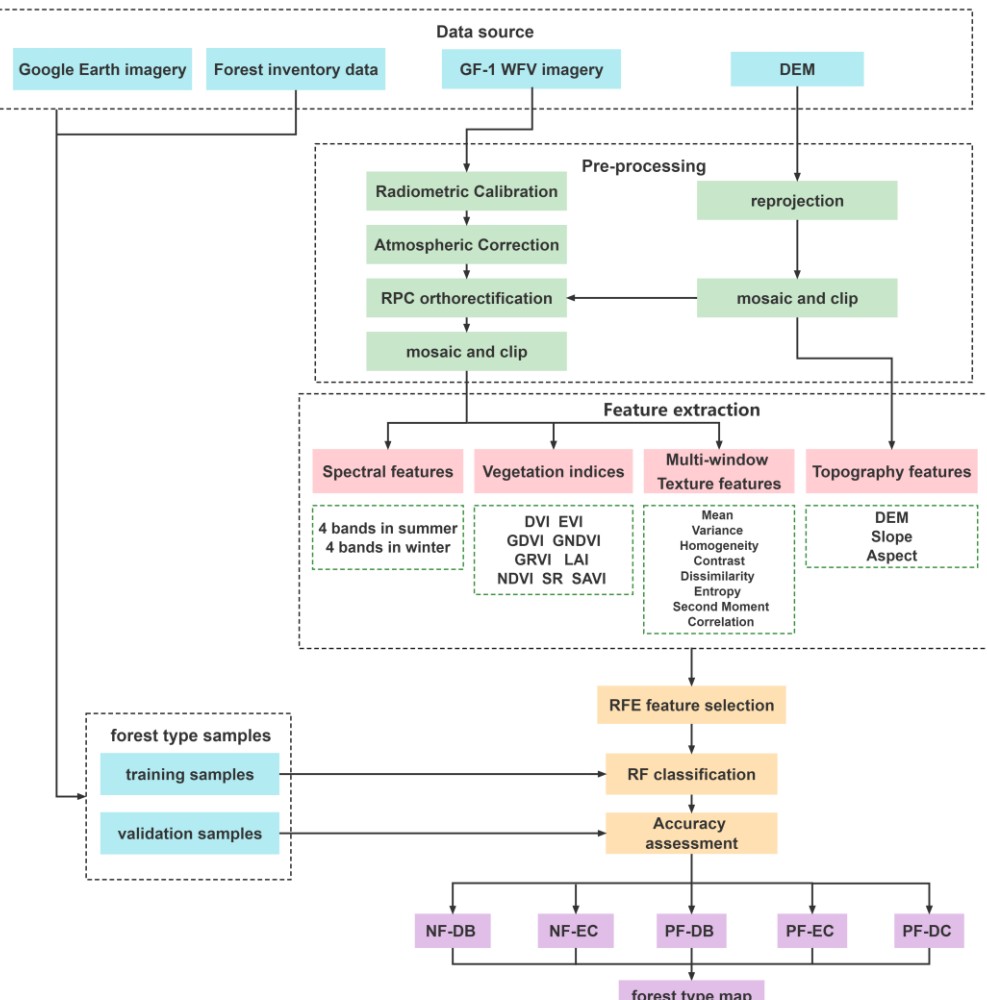

**Figure 2.** Methodological flowchart illustrating the important steps to produce the forest map.

## 3. Results

### 3.1. Feature Analysis and Selection

#### 3.1.1. Feature Analysis

We used the training samples to generate square polygons (32 m × 32 m) and counted the mean value of reflectance and mean vegetation indices of five forest types. In Figure 3, five forest types show similar spectral curves in GF-1 WFV four bands, with the lowest reflectance in the blue band and the highest in the NIR band. The reflectance is different among the five forest types.

An obvious difference can be seen in Figure 3a for the mean reflectance of PF-DB and other forest types in the GF-1 WFV summer green band and red band. The mean reflectance of the deciduous broad-leaved forest (NF-DB and PF-DB) is higher than the coniferous trees in the GF-1 WFV summer NIR band. The mean reflectance of NF-EC and PF-EC is similar in the GF-1 WFV summer bands. In Figure 3b, the differences in the reflectance among five forest types are more obvious in winter than in summer. PF-DC has the highest spectral reflectance in the blue, green and red band. PF-EC has the highest spectral reflectance in the NIR band. The reflectance of the two natural forest types decreased more than in the plantation from summer to winter.

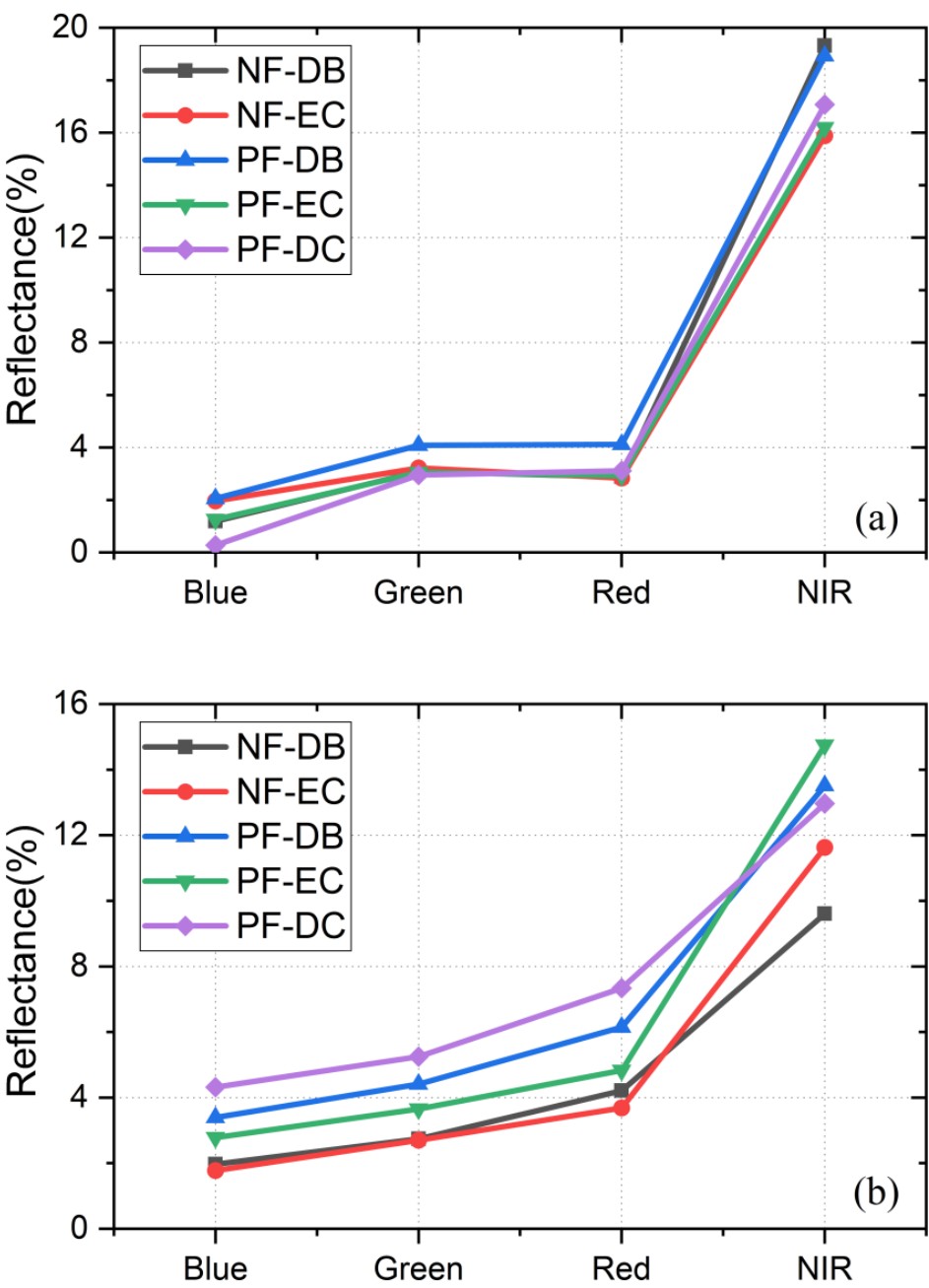

**Figure 3.** Mean reflectance of five forest types from GF-1 WFV bands: (**a**) Summer; (**b**) Winter. NF-DB: natural deciduous broad-leaved forest; NF-EC: natural evergreen coniferous forest; PF-DB: deciduous broad-leaved plantation; PF-EC: evergreen coniferous plantation; PF-DC: deciduous coniferous plantation.

As shown in Figure 4, all nine vegetation indices are higher in summer than in winter. NF-DB has the highest DVIsum, EVIsum, GDVIsum, SAVIsum and LAIsum. NF-EC has the highest GNDVIsum, NDVI sum, GRVIsum and SRsum. PF-DC has the lowest DVIsum, EVIsum, GDVIsum and LAI sum.

In winter, PC-EC has the highest DVIwin, EVIwin, GDVIwin, NDVIwin and SAVIwin. NF-EC has the highest GDVIwin, LAIwin, GRVIwin and SRwin. PF-DC has the lowest DVIwin, GDVIwin, NDVIwin and GRVIwin. PF-DB has the lowest GNDVIwin, LAIwin and GRVIwin and SRwin.

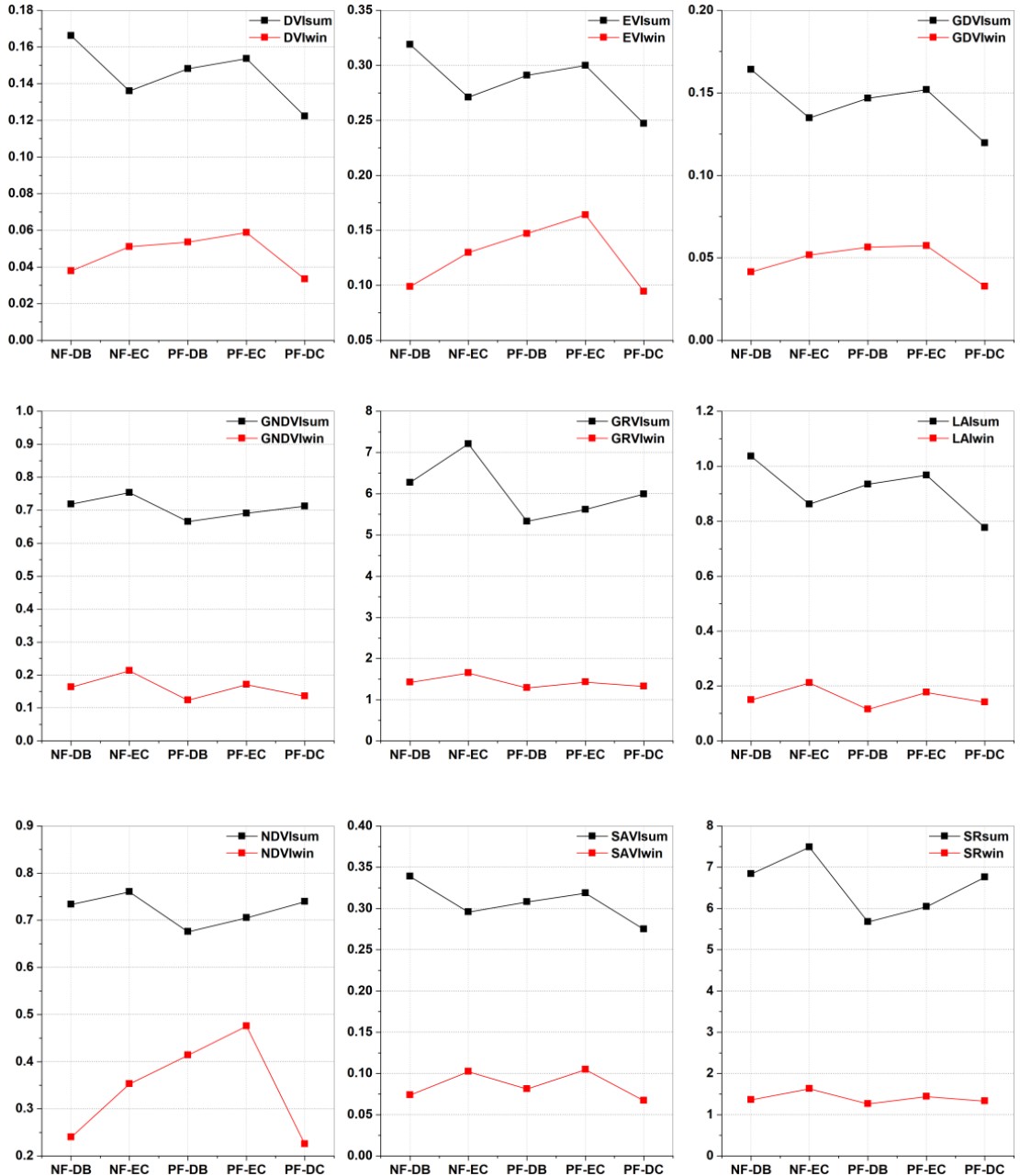

**Figure 4.** Mean vegetation indices of five forest types. Difference vegetation index (DVI); enhanced vegetation index (EVI); green difference vegetation index (GDVI); green normalized difference vegetation index (GNDVI); green ratio vegetation index (GRVI); leaf area index (LAI), normalized difference vegetation index (NDVI); simple ratio vegetation index (SR); soil-adjusted vegetation index (SAVI). The suffix sum represents VI in summer and suffix win represents VI in winter.

### 3.1.2. Feature Selection

As is shown in Figure 5, for five-forest type classification, when the number of features is 32, the classification accuracy reaches the highest with 0.8303. The importance ranking of these top 32 features is showed in Figure 6, including 8 spectral features (red color), 9 vegetation indices (green color), 13 texture features (blue color) and 2 topography features (black color). All the spectral features are included in the subset. DEM is the most vital feature for five-forest type classification. The GF-1 red band in winter follows DEM. LAI in winter and EVI in winter rank third and fourth, respectively. Most selected texture features

are from the GF-1 NIR band. Mean and entropy extracted from the GF-1 NIR band in $7 \times 7$ window rank top among texture features.

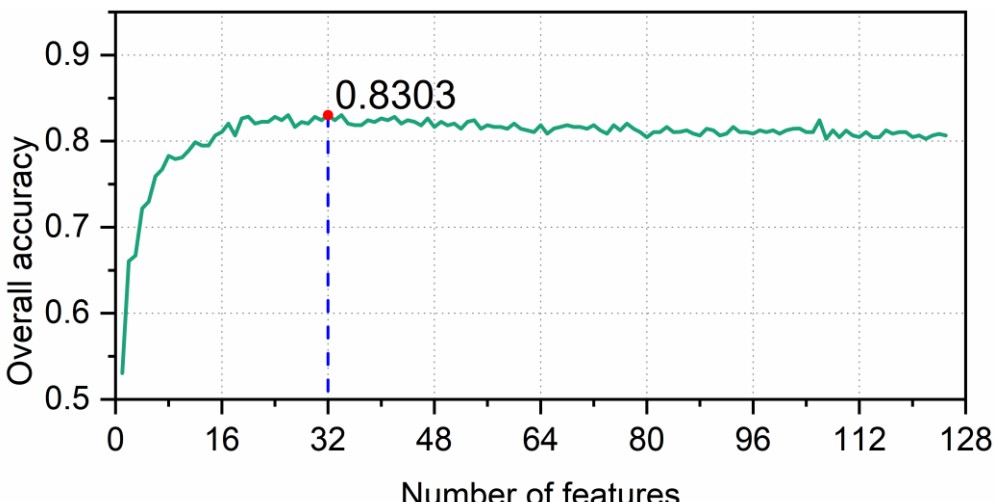

**Figure 5.** Change of overall accuracy with number of features using recursive feature elimination (RFE).

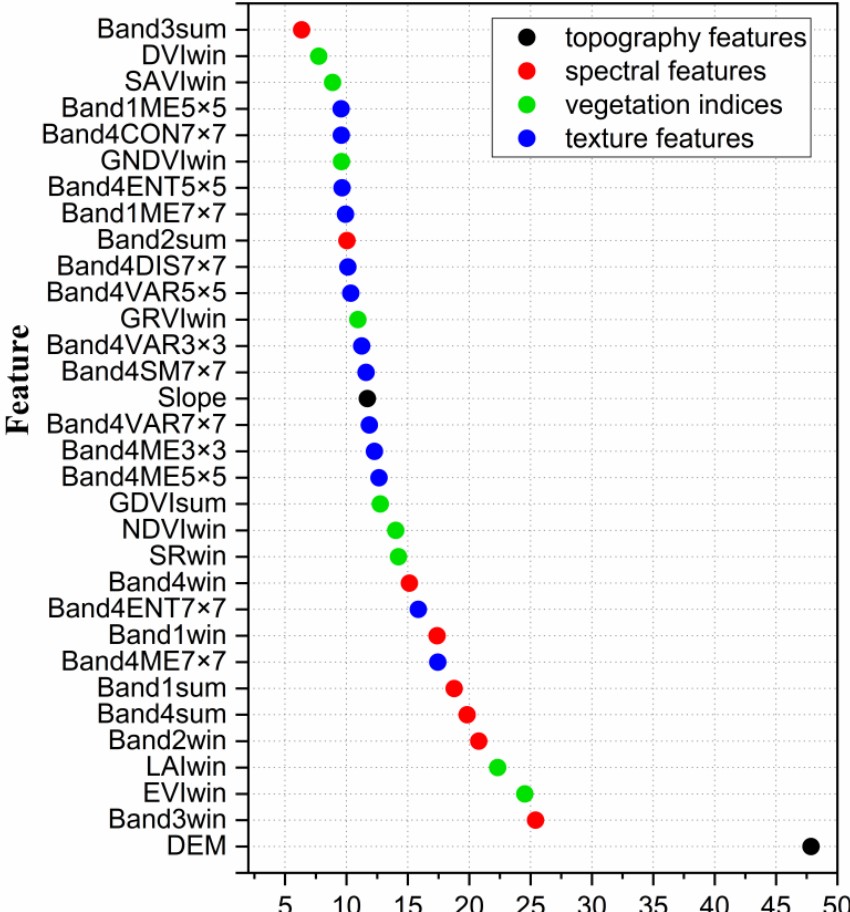

**Figure 6.** Importance of input features. Band3win: red band in winter; EVIwin: enhanced vegetation index in winter; LAIwin: leaf area index in winter; Band2win: green band in winter; Band4sum: NIR

band in summer; Band1sum: blue band in summer; Band4ME7 × 7: mean extracted from NIR band in 7 × 7 window size; Band1win: blue band in winter; Band4ENT7 × 7: entropy extracted from NIR band in 7 × 7 window size; Band4win: NIR band in winter; SRwin: simple ratio vegetation index in winter; NDVIwin: normalized difference vegetation index in winter; GDVIsum: green difference vegetation index in summer; Band4ME5 × 5: mean extracted from NIR band in 5 × 5 window size; Band4ME3 × 3: mean extracted from NIR band in 3 × 3 window size; Band4VAR7 × 7: variance extracted from NIR band in 7 × 7 window size; Band4SM7 × 7: second moment extracted from NIR band in 7 × 7 window size; Band4VAR3 × 3: variance extracted from NIR band in 3 × 3 window size; GRVIwin: green ratio vegetation index in winter; Band4VAR5 × 5: variance extracted from NIR band in 5 × 5 window size; Band4DIS7 × 7: dissimilarity extracted from NIR band in 7 × 7 window size; Band2sum: green band in summer; Band1ME7 × 7: mean extracted from blue band in 7 × 7 window size; Band4ENT5 × 5: entropy extracted from NIR band in 5 × 5 window size; GNDVIwin: green normalized difference vegetation index in winter; Band4CON7 × 7: contrast extracted from NIR band in 7 × 7 window size; Band1ME5 × 5: mean extracted from blue band in 5 × 5 window size; SAVIwin: soil-adjusted vegetation index (SAVI); DVIwin: difference vegetation index in winter; Band3sum: red band in summer.

### *3.2. RF Parameter Optimization*

As displayed in Figure 7, for five-forest type classification, when mtry ranges in the 3–17, the prediction error is lower than 0.17. When mtry = 9, the prediction error reaches the lowest value. When mtry is greater than 17, the prediction error has obvious fluctuations. Considering the RF model stability and time consumption, 16 were selected as the optimal mtry parameters for five-forest type classification in this study.

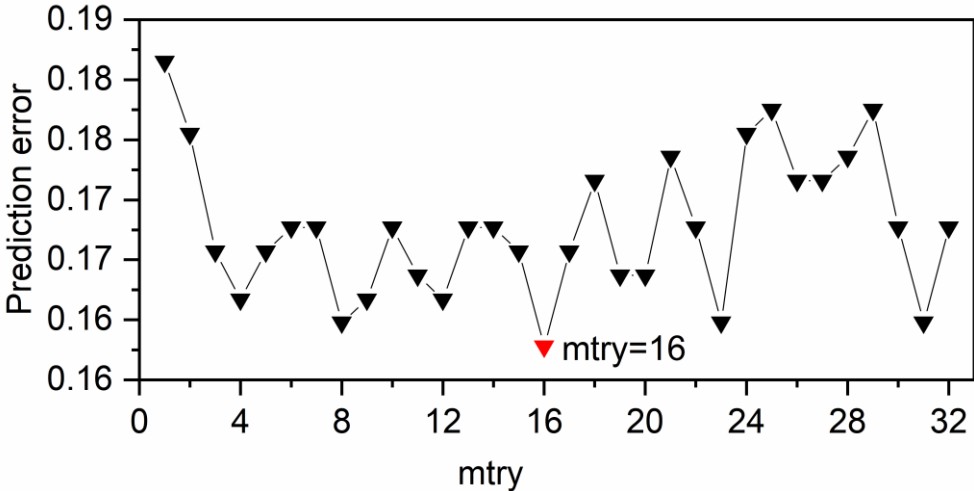

**Figure 7.** Prediction error of Random Forest model with different mtry.

The prediction error of RF models is displayed in Figure 8. When ntree ranges in the 0–300, the prediction error shows a fluctuating decrease. When ntree is greater than 300, its effect on prediction error is very limited and the prediction error tends to be stable and gains weak change. Therefore, in this study, we set ntree to 300 in for RF models.

### *3.3. Accuracy Assessment*

Table 2 shows that the classification accuracy of classification for five forest types using the optimal subset with 32 features. The overall accuracy is 87.06% with a Kappa coefficient of 0.833. We also explored accuracy of the scenario without DEM, obtaining 79.23% overall accuracy with a Kappa coefficient of 0.701.

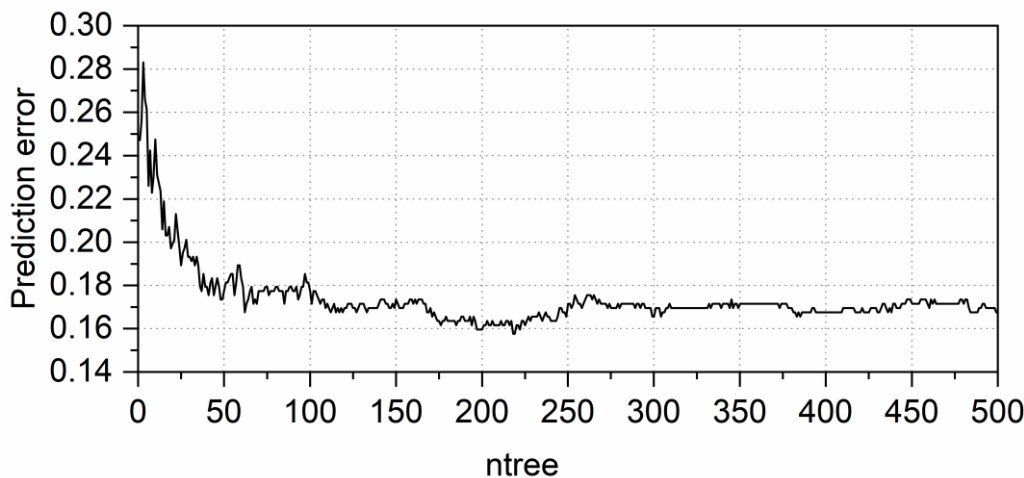

**Figure 8.** Prediction error of Random Forest model with different ntree.

**Table 2.** Confusion matrix of classification results based on RF for five-forest type classification. (UA: user's accuracy, PA: producer's accuracy, OA: overall accuracy.).

| Reference Data | Classify as | | | | | | |
|---|---|---|---|---|---|---|---|
| | **NF-DB** | **NF-EC** | **PF-DB** | **PF-EC** | **PF-DC** | **Total** | **PA (%)** |
| NF-DB | 60 | 4 | 2 | 0 | 0 | 66 | 90.91 |
| NF-EC | 2 | 20 | 0 | 5 | 0 | 27 | 74.07 |
| PF-DB | 3 | 1 | 28 | 1 | 0 | 33 | 84.85 |
| PF-EC | 2 | 4 | 1 | 38 | 0 | 45 | 84.44 |
| PF-DC | 0 | 1 | 0 | 0 | 29 | 30 | 96.67 |
| Total | 67 | 30 | 31 | 44 | 29 | 201 | - |
| UA (%) | 89.55 | 66.67 | 90.32 | 86.36 | 100.00 | - | 87.06 |
| Overall accuracy = 87.06% | | | | | | | |
| Kappa coefficient = 0.833 | | | | | | | |

From the accuracy assessment results in Table 2, NF-DB, PF-DB, PF-EC and PC-DC showed more than 84% producer's accuracy and user's accuracy, which means this study have obtained good classification results. PC-DC has obtained the highest producer's accuracy of 96.67% in the five forest types. Among five forest types, NF-EC obtained the smallest producer accuracy (74.07%). We obtained the area of the five forest types using ArcGIS 10.4 zonal statistics software. Natural forest area is 81,587 ha and plantation area is 50,509 ha. Deciduous broad-leaved forest is the dominant type of natural forest with an area of 79,504 ha. The area of natural evergreen coniferous forest is only 2083 ha, which is the smallest of the five forest types. Evergreen coniferous forest is the dominant type of plantation with 29,009 ha, accounting for 21.96% of the total forest and 57.43% of the plantation, followed by deciduous broad-leaved plantation (18,464 ha, 13.98% of the total forest and 36.56% of the plantation). Deciduous coniferous plantation area is 3036 ha, which is the smallest of the plantation. This result was applied to produce the five-forest type map of Yanqing District.

Table 3 illustrates the area of classification results for natural forest and plantation compared with inventory data in Yanqing district. The total area of forest in Yanqing is 132,096 ha based on RF classification results, compared with 112,262 ha from inventory data [73]. Area accuracy is 82.33% for total forest, 78.33% for natural forest, and 88.28% for plantation.

**Table 3.** Area of forest in Yanqing and area accuracy.

| Data Type | Classification Results (ha) | Inventory Data (ha) | Area Accuracy (%) |
|---|---|---|---|
| Forest | 132,096 | 112,262 | 82.33 |
| Natural forest | 81,587 | 67,053 | 78.33 |
| Plantation | 50,509 | 45,209 | 88.28 |

### 3.4. Forest Type Mapping

The two-forest type map and five-forest type map are displayed in Figures 9 and 10. Natural deciduous broad-leaved forest is located on mountainous area at an altitude of 800–1000 m. Most natural evergreen coniferous forest is concentrated in the Songshan National Nature Reserve, the northwest of Yanqing and others are distributed sparsely in high altitude mountains. Evergreen coniferous forest, as the dominant type of plantation, is mainly distributed in middle-elevation mountainous areas near cultivated land. Deciduous broad-leaved plantation is found in plain areas of Yanqing near water bodies and buildings. Most deciduous coniferous plantation is distributed near the Baihebao Reservoir in north Yanqing.

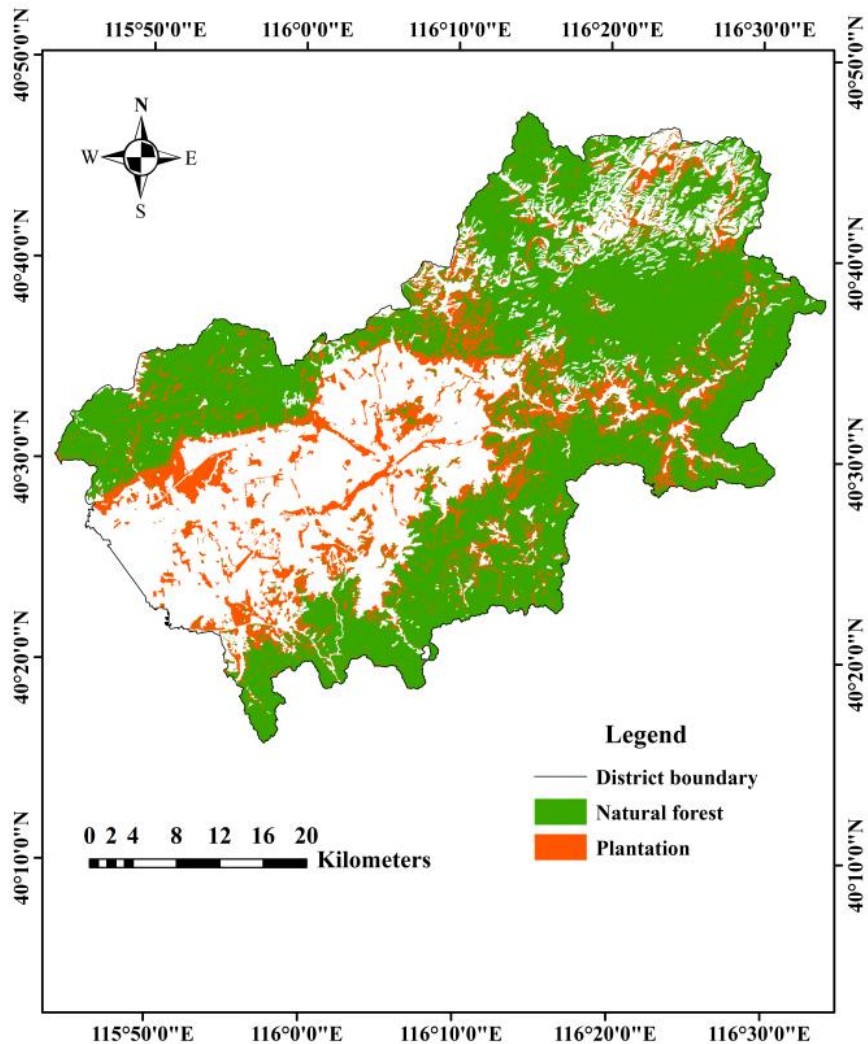

**Figure 9.** Two-forest type map of Yanqing based on random forest model.

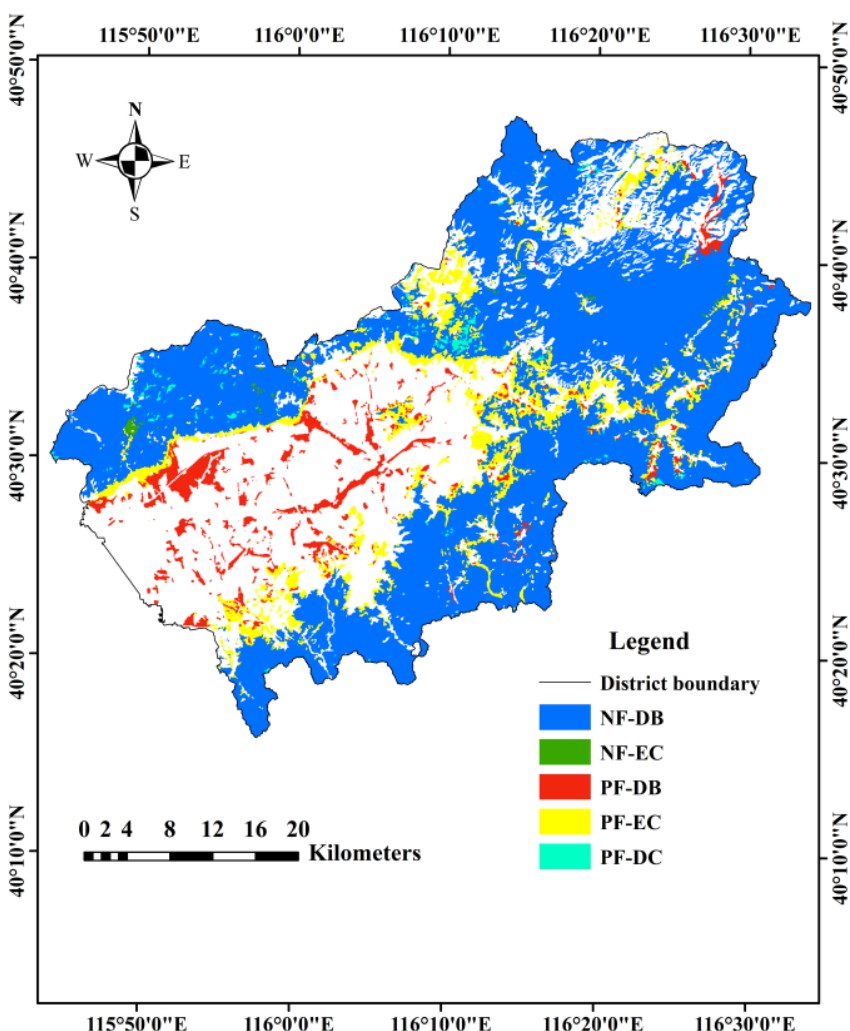

**Figure 10.** Five forest-type map of Yanqing based on random forest model. NF-DB: natural deciduous broad-leaved forest; NF-EC: natural evergreen coniferous forest; PF-DB: deciduous broad-leaved plantation; PF-EC: evergreen coniferous plantation; PF-DC: deciduous coniferous plantation.

## 4. Discussion

### 4.1. Feature Selection for RF Model

The classification results, which are derived from the optimal subset of the 32 features selected from 125 features, suggest that the RF model with RFE is a reliable approach for reducing redundant variables and improving the classifier efficiency. How to choose appropriate features is a huge challenge in land classification. By gradually eliminating the least relevant features, the RFE method not only obtains the optimal number of features for classification, but also obtains the optimal subset, which can accurately select the suitable features to improve the classification accuracy. Thus, feature selection is a necessary process to reduce input features in RF algorithm. In our study, the major target is to map forest types with five categories using the optimal subsets; thus, we used MDA to rank 32 features together. Some studies have divided the features into different groups and assessed the importance respectively. Zhang [74] obtained the feature importance of the single-scene and multi-temporal images, respectively, and in each scenario, the original spectral features, the vegetation indices and the texture features were ranked separately. Immitzer et al. [75] tested eight individual scenes and 262,143 combinations of scenes for each of the two forest strata. The study aggregated the importance of individual months and every spectral band. Grabska et al. [76] selected 12 combinations for forest tree species classification according to the variable importance and spectral-temporal patterns assessment. These research studies

aim to select the best scene for particular tree species classification, which are applicable to multi-temporal or remote sensing data with many spectral bands. Since GF-1 WFV data has four bands and the demand of mapping five categories, we did not rank the importance in each feature type, respectively. For further tree species classification, we hope to explore the best combination of multi-temporal data and spectral bands. DEM as a topographic variable ranks highest in the optimal subset. This illustrates the fact that topographic information is necessary for mapping plantations in mountainous area. This is consistent with the fact that the plantation in Yanqing District is mostly developed in the plain area, whereas the natural forest is distributed in the high-altitude area. LAI and EVI in the winter season are high on the list of feature importance. This may be due to the more accurate description of the differences in canopy and growth conditions compared with other vegetation indices.

For texture features, NIR is the most important spectral band. None of texture variables calculated by the green band from GF-1 were selected. The GF-1 NIR band contains the red edge band wavelength, which is sensitive for extracting vegetation because reflectance changes rapidly [77]. In the process of extracting texture features, the window size is an essential parameter. In this study, 7 × 7 (112 × 112 m) is the most suitable size. These results are consistent with the forest patch size in Yanqing [78]. Mean, entropy, and variance are standout textural features, while correlation and homogeneity are excluded. Mean reflects the brightness of the image. Entropy reflects the disorder degree. Variance reflects the spectral heterogeneity. The larger the VAR value, the coarser the texture. These results are similar to Liu (2020)'s study about mapping a Larch plantation in north China [79] and Li (2017)'s study about distinguishing broad-leaved forest and coniferous forest [80]. Compared to natural forest, plantation in Yanqing is established regularly, with a higher degree of order and more uniform in texture.

Synthetically, these results imply that RF model with feature selection process may have more extensive potential for application in remote sensing-based mapping.

### 4.2. Potential Application for Multi-Source Data

Previous studies mostly extracted certain plantation tree species as a separate type from other land use types, and mostly used the spectral bands and vegetation index characteristics. However, in areas with complex or rugged terrain, multiple tree species are planted. The summer spectrum of plantation and natural forest is very similar. It is difficult to distinguish plantation from natural forest using single scene images.

In this study, we used different spatial resolution data to generate four types of features. High-resolution imagery from Google Earth Pro records the temporal dynamics of forest types [81]. Additionally, the imagery provided by Google Earth Pro has displayed obvious textural characteristics (e.g., regular shapes and apparent boundaries) for plantation and was helpful for visual interpretation. So far, it has been very common to use Google Earth Pro for training and validating for typical plantation type (e.g., rubber plantations) extraction [82,83]. Forest inventory data offers accurate spatial distribution of forest with large area [58,84]. The two are combined to provide a useful platform for sample acquisition. Moreover, forest inventory data offers statistical information for accuracy assessment. The double-seasonal spectral bands and vegetation indices contribute to the classification, suggesting these variables carry information about phenology characteristics of different forest types. DEM is a key feature for extracting plantation in Yanqing. In our study, using the subset without DEM, the overall accuracy of five-forest type is less than 80%. DEM influences the afforestation and tending projects. In Yanqing, most plantation is developed in flat areas with low DEM. High DEM limits afforestation, irrigation, disease control and other measures for plantation [85]. Texture features provide discrimination of plantation and natural forest in orderliness and roughness. The integrative use of these data achieved good separability for plantation and natural forest in Yanqing.

There is limitation of the method. In our study, due to the lack of radar data, we only used optical remote sensing data, which requires cloud-free images. In large-scale regions,

it is difficult to achieve completely cloud-free images and remove cloud cover. There is still potential for studying the optimal window size and texture features for each forest types. In addition, we only used nine broadband greenness vegetation indices due to the limitation of GF-1 spectral bands. Narrowband greenness and canopy water content are not applied in this study. In future, we hope that more studies are carried out to compare different spatial resolution remote sensing images and more vegetation indices for extracting plantation in north China.

## 5. Conclusions

Plantation mapping is difficult to characterize using the traditional classification methods due to complex planting patterns and dynamic developing status. This study presented forest type mapping based on integrating multiple remote sensing images in random forest (RF) algorithm. Two-forest type and five-forest type maps in Yanqing of Beijing, north China, were produced and were assessed for accuracy. The importance measure shows DEM, EVI in winter and LAI in winter are strong contributors to plantation mapping in Yanqing. Mean, entropy and variance are vital texture features. Overall, our classification results achieved the desired accuracies, 32 features produced 87.06% accuracy for five forest types. The proposed multi-source data offers the advantage of characterizing the plantation and natural forest distribution based on the optimized feature subsets. These good results mean that the forest maps could potentially be incorporated into a range of ecological models to evaluate how plantations contribute to ecosystem services. Such extensive forest information is particularly essential for regional forest resource management.

**Author Contributions:** Conceptualization, F.W. and X.W.; methodology, F.W.; software, F.W.; validation, F.W., Y.R. and X.W.; resources, F.W. and X.W.; writing—original draft preparation, F.W.; writing—review and editing, Y.R. and X.W.; visualization, F.W.; supervision, X.W.; funding acquisition, Y.R. and X.W. All authors have read and agreed to the published version of the manuscript.

**Funding:** This research was funded by National Key R & D Program of China (2016YFC0503004) and the Special Project on National Science and Technology Basic Resources Investigation of China (2021FY100703).

**Institutional Review Board Statement:** Not applicable.

**Informed Consent Statement:** Not applicable.

**Data Availability Statement:** Not applicable.

**Conflicts of Interest:** The authors declare no conflict of interest.

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
