# Peer review of "Application of Multi-Source Data for Mapping Plantation Based on Random Forest Algorithm in North China"

_remotesensing, doi:10.3390/rs14194946_

Round 1
Reviewer 1 Report
This study aims to map tree species into several classes and also separate plantation stands from natural forests using double-date remote sensing data and site variables (DEM in particular). The novelty of this research is average and it fails to perform a comprehensive literature review. I strongly suggest the authors compare and discuss their methods, and results with similar research published recently. Still, the research seems to have some regional importance and could be considered for publication if the authors are agreeing to revise it as suggested.
Specific comments:
Page 1, line 26: Please replace increase with increasing
Page 2, line 92: Please provide some information about the GF-1 WFV imagery.
Figure 1: The figure caption should be comprehensive. Also, please define the Legend classes in your caption. The figure is of very poor quality (in terms of resolution) and misses some basic components of a map like north arrow.
Page 4, line 144: How did you generate LAI from the RGBNIR bands of the satellite data that you used?
Section 2.3.2: What feature of RF did you exactly use for variables’ importance ranking? Are you just using variable importance values after running the RF model with all the variables or you used some specific algorithm for it?
Figure 2 : The image quality is very poor plus the caption should be comprehensive and standalone.
Figure 3 : Could you clarify how you calculated these reflectance %
Figure 4: It is a great figure; however, its quality is ruining everything. Its not even legible.
Figure 6: As per Figure 6, DEM is the most important variable in your classification. I doubt this. You might want to rerun your model removing DEM and see the difference. In addition, the 32 variables you are using to build your model might be highly correlated. Its always a better approach to subjectively evaluate the variables obtained from the model. Models just deal with the numbers. Oh, I can see the report of model accuracy without DEM in later section (DEM is increasing the accuracy).
Page 10, line 261: How did you come up with the area measurement for the classes? Are you just counting the pixels?
Table 2: I would prefer to include a confusion matrix instead of just UA, and PA.
Discussion: You are missing some very important and recent works from the researchers in this field. Please consider their publications in your Introduction, Methods and Discussion sections:
Bhattarai, R., Rahimzadeh-Bajgiran, P., Weiskittel, A., Meneghini, A. and MacLean, D.A., 2021. Spruce budworm tree host species distribution and abundance mapping using multi-temporal Sentinel-1 and Sentinel-2 satellite imagery. ISPRS Journal of Photogrammetry and Remote Sensing, 172, pp.28-40.
Grabska, E., Hostert, P., Pflugmacher, D. and Ostapowicz, K., 2019. Forest stand species mapping using the Sentinel-2 time series. Remote Sensing, 11(10), p.1197.
Immitzer, M., Neuwirth, M., Böck, S., Brenner, H., Vuolo, F. and Atzberger, C., 2019. Optimal input features for tree species classification in Central Europe based on multi-temporal Sentinel-2 data. Remote Sensing, 11(22), p.2599.
Reviewer 2 Report
The manuscript entitled "Application of multi-source data for mapping plantation based on Random Forest algorithm in North China" aimed to this study explored the capability of Random Forest (RF) algorithm for the extraction and mapping of five forest types located in Yanqing, North China using Google Earth imagery, forest inventory data, GF-1 WFV images and DEM.
The manuscript is very well written; clear, and easy to understand. I think it is suitable for publication in Remote Sensing, after the authors have addressed the following comments:
1) Page 1, line 16. In this study, write the full name of GF-1 WFV.
2) Page 2, line 59. Add some references at the end of the sentence “…. Used in remote sensing research”.
3) Page 4, line 140. What does GF-1 mean? Is it GF-1 WFV?
4) Page 5, line 160. Check the numbers. I believe you mean 5 * 5 (80 * 80 m) not (90 * 90 m).
5) Page 5, line 168. Add some references at the end of the sentence “….. and much pressure of calculation”.
6) Figures 2 and 4 are important but unreadable. Reproduce them.
7) Figure 9 is also unreadable, reproduce it. In addition, Figures 9a and 9b must be placed vertically in one page.
8) Page 13, lines 340-345. Before the future directions, discuss the limitation of the method.
Round 2
Reviewer 1 Report
I can see the authors have addressed my comments. I recommend this manuscript for publication.